# Potential Therapies Targeting Metabolic Pathways in Cancer Stem Cells

**DOI:** 10.3390/cells10071772

**Published:** 2021-07-13

**Authors:** Yao-An Shen, Chang-Cyuan Chen, Bo-Jung Chen, Yu-Ting Wu, Jiun-Ru Juan, Liang-Yun Chen, Yueh-Chun Teng, Yau-Huei Wei

**Affiliations:** 1Department of Pathology, School of Medicine, College of Medicine, Taipei Medical University, Taipei 11031, Taiwan; shen1202@tmu.edu.tw (Y.-A.S.); chance03070307@gmail.com (C.-C.C.); b101106083@tmu.edu.tw (J.-R.J.); b101108037@tmu.edu.tw (L.-Y.C.); b101108068@tmu.edu.tw (Y.-C.T.); 2Graduate Institute of Clinical Medicine, College of Medicine, Taipei Medical University, Taipei 11031, Taiwan; 3International Master/Ph.D. Program in Medicine, College of Medicine, Taipei Medical University, Taipei 11031, Taiwan; 4Department of Pathology, Shuang-Ho Hospital, Taipei Medical University, New Taipei City 23561, Taiwan; b8801061@gmail.com; 5Center for Mitochondrial Medicine and Free Radical Research, Changhua Christian Hospital, Changhua City 50046, Taiwan; xxxperfume@gmail.com

**Keywords:** cancer stem cell, fatty acid metabolism, glycolysis, glutamninolysis, metabolic pathway, metabolic plasticity, mitochondrial respiration

## Abstract

Cancer stem cells (CSCs) are heterogeneous cells with stem cell-like properties that are responsible for therapeutic resistance, recurrence, and metastasis, and are the major cause for cancer treatment failure. Since CSCs have distinct metabolic characteristics that plays an important role in cancer development and progression, targeting metabolic pathways of CSCs appears to be a promising therapeutic approach for cancer treatment. Here we classify and discuss the unique metabolisms that CSCs rely on for energy production and survival, including mitochondrial respiration, glycolysis, glutaminolysis, and fatty acid metabolism. Because of metabolic plasticity, CSCs can switch between these metabolisms to acquire energy for tumor progression in different microenvironments compare to the rest of tumor bulk. Thus, we highlight the specific conditions and factors that promote or suppress CSCs properties to portray distinct metabolic phenotypes that attribute to CSCs in common cancers. Identification and characterization of the features in these metabolisms can offer new anticancer opportunities and improve the prognosis of cancer. However, the therapeutic window of metabolic inhibitors used alone or in combination may be rather narrow due to cytotoxicity to normal cells. In this review, we present current findings of potential targets in these four metabolic pathways for the development of more effective and alternative strategies to eradicate CSCs and treat cancer more effectively in the future.

## 1. Introduction

Cancer stem cells (CSCs), a particular subpopulation of cancer cells, are not only able to generate the tumor bulk [1] but are also responsible for carcinogenesis, tumor progression, resistance to chemotherapy and tumor relapse in many different types of cancer [2,3,4,5,6,7,8,9]. Conventional cancer therapies fail to eradicate cancer cells for only killing differentiated cancer cells while sparing CSCs, which leads to metastasis and recurrence of the cancer cells [10]. CSCs can compromise anticancer therapy by multiple strategies, such as expelling anticancer drug via ATP-binding cassette (ABC) transporters [11], shifting to quiescent phenotype of CSCs, remaining in the hypoxic environment [4], and even maintaining low levels of reactive oxygen species (ROS). Identification and characterization of CSCs have become the top priority in order to cure cancers. Isolation of CSC was generally based on stemness-related surface markers with fluorescence-activated cell sorting method [12], which included CD34, CD38 [13] CD133, CD44, and CD24 [14]. Presence of CD133 is representative of CSC biomarker, illustrating the stem cell properties such as multi-drug resistance, tumorigenicity, and self-renewal. Additionally, high CD44/CD24 ratio has been recognized as the potent marker of CSCs for that it indicated the proliferative capacity and the tumorigenicity of cancer [15]. Nonetheless, cell surface and cytoplasmic markers on CSCs are not static but may vary in different microenvironments [16]. Therefore, functional assays including sphere-formation assays or transplantation assays, which indicate tumorigenicity and self-renewal capacity respectively, should also be applied into the identification of CSCs whether in non-adherent serum-free conditions or in animal models [17,18]. On the other hand, CSC gene such as c-MYC and nuclear factor-erythroid 2-related factor 2 (NRf2) may contribute to distinct properties of CSC from those of normal tissue cells [19,20].

To further characterize the CSCs, heterogenic metabolic phenotypes of CSCs need detailed investigation [21]. Most tumor cells generate energy mostly by glycolysis, even in oxygen-rich environment, known as Warburg effect [22], but CSCs may preferentially use glycolysis or oxidative phosphorylation (OXPHOS) for energy production in different microenvironments. Metabolic plasticity allows CSCs to switch between different metabolic preference and continually meet high energy demand for tumor proliferation with heterogeneous nutrient supplies in the microenvironments [23,24]. For example, ovarian CSC switch between different metabolic pathways, including OXPHOS, glycolysis, glutaminolysis, fatty acid metabolism [25]. Metabolic plasticity can be one of the contributors to the challenge to eradicate CSC [26]. In this article, we have not only categorized four main metabolic pathways of CSCs, including mitochondrial respiration, glycolysis, glutaminolysis, and fatty acid metabolism, but also highlight several transcription factors and oncogenes associated with metabolic regulation in CSCs and eventually explore potential drugs targeting CSCs metabolism for future treatment of cancers.

## 2. Mitochondrial Respiration and Oxidative Phosphorylation in CSCs

### 2.1. CSCs Mainly Use OXPHOS for Energy Production

Growing shreds of evidence have substantiated that most CSCs exhibit higher mitochondrial activity and preferentially use respiration and OXPHOS as the source of energy as compared with their non-CSCs counterpart (Figure 1). It is exemplified in CSCs isolated from ovarian cancer patients by using biomarker CD44 and CD117. CD44^+^CD117^+^ (CSC) cells preferentially use the glucose to produce pyruvate fueling into the TCA cycle and ETC whereas CD44^+^CD117^−^ (non-CSC) cells exhibit a more Warburg-like profile [27]. Increased expression level of mitochondrial ATP synthase, higher mitochondrial membrane potential and accumulation of mitochondrial ROS have been observed in ovarian CSCs that make them more vulnerable to mitochondrial inhibitors. Expression profile of genes involved in glycolysis and TCA cycle show that the metabolic characteristics of CSCs are compatible with oxidative metabolism not only in ovarian CSCs, but also in CSCs of other tumor types including cervical carcinoma and pancreatic ductal adenocarcinoma [27,28,29,30]. When it comes to metabolic profile, CSCs with specialized adaptation permit metabolic switch between glycolysis and respiration to meet energy demand under physiological changes or microenvironmental stress such as quiescence state, low-oxygen concentration and nutrient deprivation [30,31]. Accumulating evidence indicates that mitochondrial function is required for self-renewal and survival of CSCs. CSCs can remain in the reversible quiescent state to survive in response to microenvironmental stress, which attributes to adaptive stress tolerance and chemoresistance property of CSCs [32,33,34]. Under glucose deprivation condition, higher dependency on OXPHOS forces quiescent CSCs to escape from metabolic stress [27,35]. The buffering capacity of mitochondria for Ca^2+^ ions has also been shown to affect the quiescence transition ability of glioblastoma with stem cell-like properties [36]. Aside from the CSCs mentioned above, many other types of CSCs rely more on mitochondrial respiration than on glycolysis, including small cell lung cancer [37], glioma, pancreatic ductal adenocarcinoma [38], lung cancer side population cells [39], and quiescent leukemia stem cells with low levels of ROS [32]. Actually, mitochondria-related properties and metabolic state are not the same across tumor types and individual. CSCs of lung cancer showed high membrane potential with low oxygen consumption and decreased intracellular levels of ATP and ROS [39]. CSCs with low mitochondrial respiration are highly dependent on glycolytic metabolism for energy requirement and for preserving stemness and chemoresistance, which are also observed in other cancers such as human glioblastoma, osteosarcoma and breast cancer [36,40,41]. Even though much attention has been paid to the causative role of specific metabolic features of CSCs in chemotherapeutic resistance and recurrence of malignant tumors, mitochondrial metabolism and physiology of CSCs in different cancer types warrant further studies. Understanding the mechanism governing the metabolic heterogeneity in CSCs can offer promising strategies for CSCs-targeting therapy in the treatment for metastasis or relapse prevention.

### 2.2. Regulation of Mitochondrial Biogenesis in CSCs

Mitochondrial biogenesis has lately come up as an important trait of CSCs. By targeting at mitochondrial biogenesis, an increase of oxygen consumption rate (OCR) and ROS was observed, which indicates an increase of the respiratory function of mitochondria [27,32,33,38,42,43,44,45,46,47,48]. It has been shown that mitochondrial biogenesis is upregulated by peroxisome proliferator-activated receptor-gamma coactivator (PGC-1α) in breast CSCs [43,49]. In pancreatic CD133^+^ CSCs, mitochondrial respiratory function is tightly regulated by PGC1-α, which is necessary for maintaining functionality and stemness of CSCs [35]. These CSCs with low glycolytic capacity are highly dependent on mitochondrial respiration and more susceptible to metformin, a Complex I-targeting inhibitor, than their non-CSCs counterpart. It was shown that MYC could negatively regulate the expression of PGC1-α in CD133^+^ CSCs, and that the overexpression of MYC enhanced glycolytic activity and subsequently conferred pancreatic CSCs greater resistance to metformin [33] (Figure 1). Moreover, mitochondrial respiration in CD133^+^ glioblastoma cells was found to be activated by IMP2 through binding and stabilizing both mitochondrial Complex I proteins and Complex IV mRNAs, thereby supporting the maintenance and propagation of CSCs [33].

### 2.3. Therapeutic Targeting OXPHOS in CSCs Metabolism

Blocking electron transport chain (ETC) is an effective strategy for inhibition of OXPHOS (Figure 1). It was found that metformin and phenformin, inhibitors of Complex I, caused mitochondrial dysfunction in pancreatic CSCs [38,50,51] (Figure 1). Metformin is a widely used anti-diabetic drug that improves the chemotherapeutic sensitivity of breast cancers to multiple anticancer drugs [52] (Figure 1). Besides, rotenone, an inhibitor of Complex I, caused incomplete electron transport and led to a decrease of ATP production in breast CSCs [53,54] (Figure 1). Moreover, NV-128, an isoflavone derivative, could block mitochondrial respiration of ovarian CSCs via inhibition of Complexes I and IV [55,56] (Figure 1). Furthermore, it was reported that diphenyleneiodonium chloride (DPI) and deferiprone (DFP) inhibit Complexes I and II in breast CSCs through blocking electron transfer of FMN and FAD-dependent enzymes and iron-sulfur clusters [57,58] (Figure 1). On the other hand, Complex III inhibitors atovaquone and antimycin A targeted at breast CSCs and lung cancer spheroids, respectively [59,60] (Figure 1). Complex V inhibitors bedaquiline and oligomycin A targeted at breast CSCs [61] (Figure 1) and glioblastoma cell lines. The effect of inhibiting the cell growth of oligomycin A is especially significant in cells treated with 2-deoxy-D-glucose (2-DG) [62] (Figure 1). In addition to the ETC inhibitors mentioned above, many other chemicals targeting at different pathways in mitochondria can also reduce the function and efficiency of OXPHOS. As an example, salinomycin (procoxacin), a polyether potassium ionophore antibiotic, targets breast and nasopharyngeal CSCs through suppression of the Wnt/β-catenin pathway [34,63,64] (Figure 1). The anticancer effect of salinomycin was observed when it was used alone or combined with other drugs (5-fluorouracil and oxaliplatin) [65] (Figure 1). Epigallocatechin-3-gallate (EGCG), which blocks the Wnt/β-catenin signaling pathway, showed anticancer effect against colorectal CSCs [66,67] (Figure 1). Furthermore, ABT-263 and quercetin reduced the efficiency of OXPHOS in acute myeloid leukemia stem cells and gastric CSCs by regulating the expression of Bcl-2 [32,68] (Figure 1). Additionally, XCT-790 is an inhibitor of the ERRA/PGC-1α pathway and affects mitochondrial biogenesis in breast CSCs [43,69] (Figure 1). Moreover, linamarase, linamarin and glucose oxidase (lis/lin/GO), an inducing-system of mitophagy, leads to tumor regression by the combination with cyanide and oxidative stress. This treatment impairs the OXPHOS system of human malignant tumors [70] (Figure 1). Besides, it was reported that doxycycline, azithromycin, and tigecycline could impair mitochondrial biogenesis, respectively, by blocking protein synthesis in breast CSCs and acute myeloid leukemia stem cells [71,72,73,74] (Figure 1). Finally, considering the metabolic plasticity of CSCs, BRD4 inhibitor JQ-1 enforce dependence on OXPHOS in pancreatic CSCs and decease the intermediate glycolytic/respiratory phenotype, which enhance the efficacy of metformin as an anti-mitochondrial respiration regimen [38]. This enlightened us with the potential of the combinatorial therapies that impinge on adaptive metabolism of CSCs to subvert drug resistance and enhance the therapeutic efficacy, which is a clinically feasible approach. 

## 3. Glycolytic Switch in CSCs

Cancer cells generate ATP mostly by aerobic glycolysis, even when oxygen supply is sufficient, which is well-known as the Warburg effect. As for the CSCs, recent evidence has shown that CSCs or cancer stem-like cells have higher glucose uptake and lactate production than the non-CSCs counterparts [25], both in vitro and in vivo. This phenomenon has been observed in many types of cancer, such as breast cancer, lung cancer, colon cancer, osteosarcoma, glioblastoma, and even in ovarian cancer cells with sphere-forming properties [36,40,41,75,76]. Moreover, in the reprogramming process of induced pluripotent stem cells (iPSCs), glycolytic switch occurs prior to the expression of pluripotency markers, which is in line with the observed glycolytic switch in the process of acquiring the stemness properties of CSCs [77,78].

### 3.1. Glycolysis in CSCs

CSCs can increase the glucose intake for glycolysis by upregulating the expression of glucose transporters GLUT1 and GLUT3 (Figure 2). Aldehyde dehydrogenase (ALDH) can increase the amount of GLUT1, which in turn upregulates glycolysis and contributes to the maintenance of CSC stemness, especially in endometrial CSCs [79]. Besides, the upregulation of enzymes participating in glycolysis is evident in CSCs, including 6-phosphofructo-2-kinase/fructose-2,6-biphosphatase (PFKFB), hexokinase 2 (HK2), pyruvate kinase isozyme M2 (PKM2), glucose-6-phosphate dehydrogenase (G6PD), and lactate dehydrogenase (LDH) (Figure 2). In breast CSCs, enzymes related to glycolysis, such as PKM2, LDH and G6PD, are upregulated [40]. On the other hand, upon treatment of the breast CSCs with 2-DG, which is a glucose analogue that inhibits HK2, the proliferation rate of CSCs drops significantly [40] (Figure 2). Loss of fructose-1,6-biphosphatase (FBP1), which is an enzyme that promotes gluconeogenesis, inhibited mitochondrial function but enhanced glycolysis and stemness properties in basal-like breast cancer cells [80]. On the contrary, overexpression of FBP1 led to the upregulation of gluconeogenesis and downregulation of glycolysis, which decreased tumor spheroid formation and the number of cancer cells with stemness properties [81]. In glioblastoma stem cells, hypoxia-inducible factor (HIF)-1α, HIF-2α [82], GLUT3 [83] and glycolysis-related enzymes, such as pyruvate dehydrogenase kinase (PDK) 1, PFKFB4, and PKM2, were upregulated [84] and the stemness properties of CSCs enhanced. Knockdown of PFKFB4 triggered apoptosis of CSCs but overexpression of PFKFB4 was associated with a shorter survival of patients with glioblastoma [84]. As for the mesenchymal glioblastoma stem cells, it was found that inhibition of the ALDH1A3 attenuates glycolysis [85]. In the CSCs of nasopharyngeal carcinoma, it was found that glycolysis was increased by the upregulation of GLUT1 and glycolysis-related enzymes HK2 and PDK and the downregulation of gluconeogenic enzymes G6PC and PEPCK [86].

Transcription factors also promote glycolysis via upregulation of some glycolytic enzymes. HIF-1α upregulates GLUT, HK2, pyruvate kinase (PK) and LDHA whereas downregulates pyruvate dehydrogenase (PDH) by PDK1-mediated phosphorylation to block oxidative metabolism [87]. A higher level of expression of HIF-1α was observed in myeloid ecotropic viral integration site 1 in order to promote glycolysis [88] (Figure 2). On the other hand, MYC also directly triggers the expression of multiple glycolysis-related enzymes, such as GLUT1, HK2, PKM2, LDHA, and enolase 1 [89] (Figure 2). Overexpression of MYC enhanced the stemness properties and glycolysis in nasopharyngeal cancer, hepatocellular carcinomas and breast cancer [23,90]. MYC has been emerged as an important factor for the maintenance of CSCs traits, and its expression contributes to drug resistance in pancreatic and colon CSCs [27,32,33,38,42,43,44,45,46,47,48,91]. Thus, targeting MYC pathways may subvert therapeutic resistance. Finally, OCT4 could also initiate the transcription of HK2 and PKM2 to enhance glycolysis in mouse embryonic stem cells [92] (Figure 2). 

Oncogene-mediated signaling is involved in the switch to glycolysis. In pancreatic CSCs, knockdown of oncogenic KRAS affected glucose uptake and glycolytic flux, which impeded the stemness and chemoresistant properties [93,94,95]. KRAS not only drives glycolysis but also promotes pentose phosphate pathway (PPP) to produce precursors for the biosynthesis of nucleotides [93,94,95] (Figure 2). Interestingly, inflammatory cytokines may be associated with metabolic hallmarks (e.g., glycolysis) of CSCs. Overexpression of IL-8 was observed in colon and lung CSCs, which indicates the importance of cytokines in the development of properties of CSCs via upregulation of GLUT3 and glucosamine fructose-6-phosphate aminotransferase [96].

### 3.2. Therapy Targeting Glycolysis in CSCs

Previous studies have shown that downregulation of certain glucose transporters and glycolytic enzymes, such as GLUT1–4, HK1-2, PKM2, and LDH, can inhibit glycolysis in CSCs [97]. However, direct glycolysis inhibition has shown little success. Glucose transport inhibitors such as WZB117, fasentin, phloretin and silybin/silibinin, which are currently in phase I/II clinical trials for advanced hepatocellular carcinoma and prostate cancer, led to side effects because of the ubiquitous expression of glucose transporters [98] (Figure 2). The HK2 inhibitor, lonidamine, has shown limited improvement in survival in breast, lung and ovarian cancers [99,100,101,102] (Figure 2). 2-DG was recently tested in a phase II/III clinical trial for prostate cancer (NCT00633087) but was terminated for slow accrual (Figure 2). Since signaling pathways also affect glycolysis, targeting them may also hinder glucose metabolism. Bromodomain and extraterminal motif (BET) proteins inhibitors suppress tumor growth via downregulation of MYC [103,104,105] (Figure 2). Under hypoxic conditions, glycolysis is mainly activated in tumor cells, but OXPHOS may be retained to generate energy required for CSCs trait.

## 4. Glutamine Metabolism in CSCs

Glutamine metabolism provides precursors for the biosynthesis of nucleic acids, lipids, amino acids, and glutathione (GSH), which is an antioxidant required to maintain redox homeostasis in normal tissue cells [106]. Studies showed that cancer stem cells tend to show high expression of MYC gene, and vulnerable to glutamine deprivation due to its high proliferation and maintaining GSH and ROS level [107,108].

Glutaminolysis transforms glutamine through a series of enzymes into metabolites that may feed into the TCA cycle. First, glutamine in tissue cells is obtained from blood circulation through alanine-serine-cysteine transporter (ASCT2) or solute carrier family 1 member 5 (SLC1A5) and glutamine is converted by glutaminase (GLS) 1 and GLS2 into glutamate. Glutamate is then metabolized by glutamate dehydrogenase (GLUD) or transaminase to α-ketoglutarate (α-KG), which serves in the TCA cycle as an anaplerotic substrate [109] (Figure 3). The glutamine metabolism in CSCs is critical for the proliferation and survival of epithelial CSCs since the GSH produced from glutamine metabolism can neutralize excessive amounts of ROS (Figure 3). In addition, epithelial CSCs have improved metabolic versatility to use various carbon sources such as glutamine through the TCA cycle [110] (Figure 3).

### 4.1. Glutaminolysis in CSCs

Glutaminolysis is inescapably related to metabolic regulators such as MYC in CSCs [19]. The oncogene c-MYC induces overexpression of glutamine importers such as ASCT2, which enhanced glutamine uptake, induced glutamine synthetase and GLS transcription in CSCs [111,112] (Figure 3). MYC-dependent metabolic reprogramming is linked to CD44 variant-dependent redox stress regulation in CSCs [113]. It has been shown that MYC plays a cooperative role in the control of phenotypes and radioresistance following glutamine deficiency in neuroblastoma CSCs [107]. Furthermore, in metformin-resistant SW620 CSCs, MYC knockdown induced suppression of CSCs and enhanced suppressive effect of metformin and glutaminase C inhibitor on CSCs [114] (Figure 3). It was found that MYC is overexpressed in TNBC CSCs, where it promoted the self-renewal ability and chemoresistance of CSCs [115].

Assays of glycolysis and glutamine transporters indicate that ovarian clear cell carcinoma (CCC) displayed low glycolytic flux, which may be related to the high glutaminolysis and its CSC-like properties [116]. Especially, higher expression levels of ASCT2 and GLS were observed in CSCs, which implicated that ASCT2 and GLS are important for the proliferation and maintenance of the redox state in CSCs [114]. (Figure 3). CSCs of both HT29 and SW620 cancer cells showed a higher expression of ASCT2 [114]. Moreover, it was also found that CSCs of a patient with radioresistant prostate cancer had a high glutamine demand due to overexpression of GLS induced by the transcription factor MYC [117].

Nuclear factor-erythroid 2-related factor 2 (Nrf2) is a transcriptional regulator element that binds to anti-oxidative genes promoter to maintain cellular redox status by promoting antioxidants such as GSH generating enzyme, and gamma-glutamyl cysteine synthetase (GCS) [118,119] (Figure 3). Recent research has shown that Nrf2 signaling is involved in the CSC-like properties of many types of cancer stem cells. The biosynthesis of glutathione from glutamine, for example, has been related to the enrichment and drug resistance of breast CSCs [20]. Moreover, Nrf2 knockdown impaired the ability of glioma stem cells to self-renew [120]. Besides, it was discovered that Nrf2 signaling is activated in spheroid culture of breast cancer [121] and that Nrf2 expression is greatly increased in colon CSCs [122]. As a result, high Nrf2 activity in these CSCs-enriched systems caused anticancer drug resistance and facilitated the survival of CSCs.

### 4.2. Therapy Targeting Glutaminolysis in CSCs

Accumulating evidence have shown that glutamine depletion or inhibition of the crucial regulators of glutaminolysis, such as GLS and the transcription factor MYC are promising strategies to target glutaminolysis of CSCs (Figure 3). Multiple drugs have shown their therapeutic effects by targeting GLS1 in glutaminolysis of CSCs. Bis-2-(5-phenylacetamido-1,3,4-thiadiazol-2-yl) ethyl sulfide (BPTES) and telaglenastat (CB-839) are the allosteric inhibitors of GLS1 (Figure 3). The potency and kinetic behavior of CB-839 are different from those of BPTES. In vitro clonogenicity of glioblastoma stem-like cells (GSCs) neurosphere colonies have been effectively decreased by compounds 968 and CB-839. The above results of both inhibitors were extremely GLS-specific, as sensitivity to treatment was strongly linked to the protein expression of GLS [123]. Furthermore, the combination of metformin and compound 968 improved CSCs suppression in colorectal CSCs [114]. CB-839 inhibition of GLS resulted in substantial radiosensitization in prostate cancer cell cultures, lowering the stemness properties of prostate cancer cells [118]. Zaprinast, a phosphodiesterase 5 inhibitor and asthma drug, was recently discovered to be an inhibitor of the enzyme glutaminase (GLS) [124]. Interestingly, it is demonstrated that radiology treatment with zaprinast and BPTES could effectively sensitize pancreatic CSCs and increase apoptosis through intracellular ROS accumulation [108] (Figure 3). The above findings indicate that inhibition of glutamine metabolism is a potential therapeutic approach for increasing the radiosensitivity by disrupting redox balance [108]. Lastly, azaserine, acivicin, and 6-diazo-5-oxo-L-norleucine (DON) are analogs of glutamine that inactivate GLS1. These compounds have shown the suppression of the development of a variety of tumors and have exhibited their activity in several clinical studies [125] (Figure 3). Evidence revealed that the apoptotic population in breast cancer stem cells was significantly increased under the treatment of DON [126]. Even though DON has been commonly used in studies on the physiological functions of GLS, its intrinsic lack of selectivity and low potency has still impeded its use in cancer therapy [127].

Nrf2 inhibitors are another potential therapy for CSCs because they resulted in increase of intracellular ROS, increased radio-sensitivity of spheres, and reduced cell growth [128]. Hinokitiol, a natural bioactive compound of aromatic tropolone, has recently been discovered to suppress Nrf2 expression in glioma stem cells (GSCs), resulting in the decrease of self-renewal capacity, migration, invasiveness, and colony-forming ability of the CSCs [129] (Figure 3). Besides, treatment with chestnut leaf extract blocked the activation of the Nrf2 antioxidant protection in MCF-7-derived breast CSCs. It inhibited sphere cell development and increased cell resistance to the antitumor drug paclitaxel [130] (Figure 3). In conclusion, current therapies are mostly targeting GLS1 and Nrf2 in various types of CSCs and the combination therapy targeting glutaminolysis may have therapeutic potential to improve the clinical outcome of cancer patients.

## 5. Fatty Acid Metabolism in CSCs

### 5.1. The Importance of Fatty Acid Metabolism in CSCs

Cells acquire abundant energy from fatty acids metabolism, and recent studies have demonstrated the contribution of lipid metabolism to the regulation of CSCs, including alterations in lipid catabolism, β-oxidation of fatty acids, de novo lipogenesis, and lipid desaturation (Figure 4). Alterations of lipid metabolism support the energy need and biomass generation, and activation of several significant oncogenic signaling pathways in CSCs [131]. Fatty acid metabolism is delicately controlled by fatty acid synthesis (FAS) and fatty acid oxidation (FAO) in CSCs. FAS is an anabolic way to convert acetyl-CoA to malonyl-CoA, which is required for cell proliferation; and FAO is a catabolic pathway to use fatty acids to generate acetyl-CoA for the production of ATP [25]. Lipid metabolism plays an important role in regulating the development of CSCs and their chemoresistance. It was found that inhibiting JAK and STAT3 by the pan-JAK small-molecule inhibitor AZD1480 blocked breast CSCs (BCSCs) progression and cancer chemoresistance [132] (Figure 4). Aside from enhancing human BCSC stemness, JAK/STAT3 also participates in the expression of various genes involved in lipid metabolism [132]. For example, STAT regulates the gene expression of carnitine palmitoyltransferase 1B (CPT1B), a key enzyme for β-oxidation of fatty acids, by directly binding to its gene promoter, and this pathway is also upregulated by mammary adipocyte-derived leptin [133] (Figure 4). Lipid metabolism is also associated with the incidence and prognosis of colon cancer. Since highly proliferating cells require more major lipid constituents of membranes, CSCs accumulate unsaturated fatty acids as lipid precursors, such as monounsaturated fatty acids (MUFAs). Lipid desaturation is carried out mainly by stearoyl-CoA desaturase 1 (SCD1), which transforms saturated fatty acids (SFAs) to MUFAs, and SCD1 has been suggested to be a target for selective elimination of CSCs (Figure 4). A positive correlation was found between the level of SCD1 expression and clinical staging of colorectal cancer patients and the expression levels of genes related to CSCs properties [133]. Moreover, SCD1 can subsequently increase MUFAs and in turn upregulate Wnt and Notch signaling pathways in CSCs [133].

Additionally, specific lipid compositions are critical for the maintenance of CSCs. Some phospholipids, such as sphingomyelin, which is essential for lipid raft formation, may mediate other cell signaling pathways that promote the maintenance of colon CSCs [134,135]. In ovarian CD133^+^ CSCs that grow as spheroids, increase of unsaturated fatty acids content enhances nuclear factor kappa B (NFκB) activity, thus boosting the expression of SCD1, which then supports the stemness of cancer cells (Figure 4). It presents a feedforward loop that unsaturated fatty acids increase SCD1 via NFκB, leading to elevated production of unsaturated fatty acids [136] (Figure 4). Several enzymes involved in lipid metabolism also play important roles in conferring the stemness properties of cancer cells. For example, arachidonic acid 5-lipoxygenase (ALOX5), which is involved in the generation of leukotrienes (LT) from arachidonic acid (AA), and acyl-CoA synthetase very-long-chain 3 (ACSVL3/SLC27A3), which are key enzymes required for the formation of fatty acyl-CoA, was found to be required for the self-renewal of CSCs in glioblastoma [25] (Figure 4).

### 5.2. Therapy Targeting Fatty Acid Metabolism in CSCs

Current studies have shown that the anticancer therapeutic targets are usually enzymes that linked to fatty acid metabolism in CSCs. In GSCs, fatty acid synthase (FASN) can be upregulated by the activation of the PI3K/Akt kinase pathway [137], whereas the inhibition of FASN with cerulenin, a fungal metabolite, attenuates the GSC stemness properties by downregulating the de novo lipogenesis [137] (Figure 4). In BCSCs, treating peroxisome proliferator-activated receptor-γ (PPARγ) with its antagonist GW9662 inhibited the expression of stemness genes and lipogenic enzymes such as FASN and ATP citrate lyase (ACLY) [138], and acetyl-CoA carboxylase (ACC) inhibitor soraphen similarly reduced sphere formation potential [139] (Figure 4). As for the abovementioned enzyme SCD1, blocking SCD1 in BCSCs by curcumin has also been observed to downregulate the expression of the genes related to stemness properties [140] (Figure 4). Moreover, by using the inhibitors CAY10566 and SC-26196, the generation of MUFA from SFAs was reduced in ovarian CSCs [141] (Figure 4). Accordingly, targeting essential enzymes and factors involved in fatty acid metabolism can be one potential therapeutic strategy to eradicate CSCs and cure cancers [131].

## 6. Conclusions

Depending on the environmental stimuli and nutrient supply, CSCs can switch between various metabolic phenotypes. Under normoxic conditions, CSCs generate most of the ATP they need from OXOPHOS and β-oxidation of fatty acids [142]. On the other hand, under hypoxic conditions, glycolysis is upregulated in CSCs by enhancing the expression of essential glycolytic enzymes and GLUT [143]. Apart from generating energy, this glycolytic switch plays a vital role in promoting the stemness properties of CSCs [77,78]. Glutaminolysis is responsible for the supply of precursors for the biosynthesis of macromolecules and glutathione in CSCs. Fatty acid metabolism is reprogrammed to maintain the integrity of the cell membrane of CSCs. Accumulated evidence has suggested that the microenvironment condition governing metabolic reprogramming in CSCs is essential for metastasis and recurrence of cancer cells. Therefore, targeting the metabolism of CSCs has provided insights in developing new therapeutic drugs to overcome the insufficiency of conventional anticancer drugs in eliminating cancer cells.

A wealth of studies has unraveled the essential roles of specific metabolic enzymes, transporters and even signaling pathways in CSCs and their possibility as new targets of anticancer drugs. Nonetheless, current metabolism-targeting drugs still encountered low efficacy due to the lack of a comprehensive understanding of the metabolic heterogeneity and plasticity of CSCs. Therefore, metabolic pathways of CSCs should be more clearly elucidated and the mechanism of metabolic switch should be unraveled. Moreover, novel metabolism-targeted anticancer drugs should not be confined to only narrowly targeting a single metabolic pathway. The development of drugs targeting multiple metabolic pathways simultaneously is warranted to overcome the metabolic switch of CSCs. For instance, we can apply dual inhibition of OXPHOS and glycolysis, such as metformin in combination with a PI3K inhibitor [144] or a BET inhibitor, JQ-1 [38]. Additionally, the metabolism-targeting drugs can be used as an add-on treatment of current cytotoxic regimens. However, the synergism and efficacy of the combined therapy warrants further investigation. In order to eradicate cancer cells, the top priority is to discover biomarkers of CSCs to specifically characterize and distinguish CSCs from the tumor bulk. Also, distinctions between properties of CSCs and those of normal stem cells should also be discovered so as to eradicate cancer cells without damaging normal tissue cells. The therapeutic window may be rather narrow for some metabolic inhibitors due to toxicity to normal tissue cells. Once metabolic pathways of CSCs are clearly elucidated, novel metabolic therapeutic agents can be developed to couple with current cytotoxic regimens to eliminate cancer cells more efficiently, and thereby prevent recurrence and metastasis of cancer cells and eventually cure cancer patients.

Glucose is first imported into CSCs through a glucose transporter (GLUT) and then metabolized by multiple glycolytic enzymes to generate ATP. Besides, glycolysis also provides precursors for biosynthesis of amino acids and nucleotides. Several factors, such as hypoxia-inducible factor (HIF), MYC, OCT4, MEIS1 affect the flux of the glycolytic pathway. HIF promotes glycolysis by upregulation of GLUT and glycolytic enzymes, which are also targets of some anticancer drugs, such as WZB117, fasentin, phlorstin, silybin/silibinin, 2-deoxyglucose (2-DG), ionidamine, and the BET inhibitor.

Glutamine is obtained by tissue cells through ASCT2 (SLC1A5) and transformed into glutamate through glutaminase GLS1 and GLS2. Glutamate is then metabolized by glutamate dehydrogenase (GLUD) or by transaminase to generate α-KG, which serves as an anaplerotic substrate in the TCA cycle. Oncogene c-MYC induces overexpression of glutamine importers ASCT2 and GLS transcription. Nuclear factor-erythroid 2-related factor 2 (Nrf2) is a transcriptional regulator that binds to anti-oxidative genes promoter to maintain cellular redox status by promoting the expression of antioxidant enzymes such as gamma-glutamyl cysteine synthetase (GCS), an enzyme that generates GSH. Multiple therapeutic targets for GLS in CSCs glutaminolysis. Bis-2-(5-phenylacetamido-1,3,4-thiadiazol-2-yl) ethyl sulfide (BPTES) and CB-839 are allosteric inhibitors of GLS1, which inhibits the influx of glutamine derivatives into the TCA cycle. Zaprinast is a phosphodiesterase 5 inhibitor and an asthma drug that could effectively sensitize cancer stem cells and increase apoptosis through intracellular ROS accumulation in combination with zaprinast and BPTES. Lastly, azaserine, acivicin, and 6-diazo-5-oxo-L-norleucine (DON) are glutamine analogs that inhibit the activity of GLS1. Nrf2 inhibitors are another potential therapeutic agent for CSCs. Hinokitiol has recently been discovered to suppress Nrf2 expression in glioma CSCs, resulting in decreased self-renewal capacity, migration, invasiveness, and colony-forming ability. In addition, treatment with chestnut leaf extract blocked the activation of the Nrf2 antioxidant protection in MCF-7-derived breast CSCs. Combination therapy of *Castanea crenata* leaf extract with paclitaxel could dramatically increase death of cancer cells.

Lipid metabolism plays an important role in regulating the stemness properties of CSCs. In the pathway of fatty acid metabolism, acyl-CoA synthetase very-long-chain 3 (ACSVL3/SLC27A3) promotes the conversion of fatty acyl adenylate to fatty acyl CoA, which subsequently leads to β-oxidation of fatty acids and confers the stemness properties to CSCs. The generation of fatty acyl carnitine is supported by carnitine palmitoyltransferase 1B (CPT1B), which is upregulated by leptin-induced JAK/STAT3 signaling. JAK/STAT3 signaling can be repressed by AZD1480, a pan-JAK small molecule inhibitor, which may also affect the generation of CPT1B. Lipid desaturation has also been suggested to enhance CSCs maintenance, which is critically mediated by an increase of the stearoyl-CoA desaturase 1 (SCD1) activity. The enhanced expression of SCD1 leads to the transformation of saturated fatty acids (SFAs) to monounsaturated fatty acids (MUFAs), which subsequently promotes the activity of nuclear factor kappa B (NFκB). There is a feedforward loop that unsaturated fatty acids increase SCD1 via NFκB. The addition of curcumin blocks the expression of SCD1, which at last exerts the effect similar to CAY10566 and SC-26196 that influences the conversion of SFAs to MUFAs. Lipid metabolism enzymes such as arachidonic acid 5-lipoxygenase (ALOX5) also support the maintenance of stemness of CSCs by increasing the generation of leukotrienes (LT) from arachidonic acid (AA). Anticancer therapies targeting fatty acid metabolism may inhibit enzymes to modulate the stemness property of CSCs. The abovementioned pathways including fatty acid synthesis can be inhibited by cerulenin, soraphen and by blocking PPARγ signaling with GW9662, respectively.

## Figures and Tables

**Figure 1 cells-10-01772-f001:**
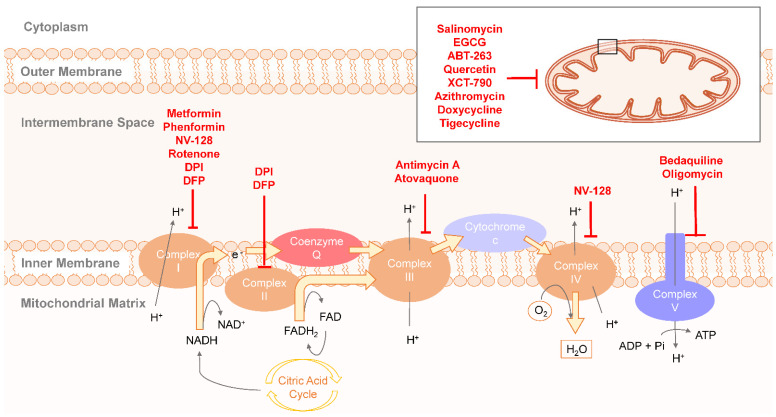
Electron transport chain and OXPHOS as the target of CSC metabolism. Many drugs have been used in targeting cancer stem cells (CSCs). Metformin, a well-known drug, is the inhibitor of Complex I. Phenformin, NV-128, and rotenone also block the electron transfer of Complex I. Diphenyleneiodonium chloride (DPI) and deferiprone (DFP) are the inhibitors of Complexes I and II. Furthermore, antimycin A and atovaquone target the ETC by targeting Complex III. In addition, bedaquiline, and oligomycin show their inhibitory effects against CSCs through blocking Complex V. For mitochondrial biogenesis, salinomycin, epigallocatechin-3-gallate (EGCG), ABT-263, quercetin, XCT-790, azithromycin, doxycycline, tigecycline are all effective chemical compounds or drugs that can regulate mitochondrial biogenesis.

**Figure 2 cells-10-01772-f002:**
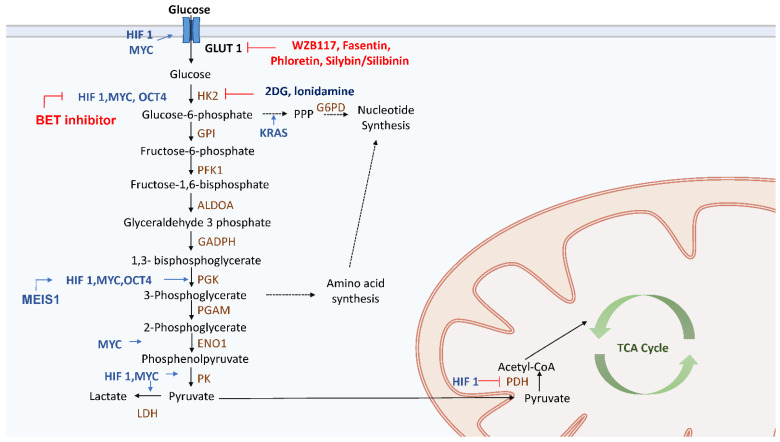
Glycolytic pathway and targets of anticancer drugs in CSCs. Glucose is first imported into CSCs through a glucose transporter (GLUT) and then metabolized by multiple glycolytic enzymes to generate ATP. Besides, glycolysis also provides precursors for biosynthesis of amino acids and nucleotides. Several factors, such as hypoxia-inducible factor (HIF), MYC, OCT4, MEIS1 affect the flux of the glycolytic pathway. HIF promotes glycolysis by upregulation of GLUT and glycolytic enzymes, which are also targets of some anticancer drugs, such as WZB117, fasentin, phlorstin, silybin/silibinin, 2-deoxyglucose (2-DG), ionidamine, and the BET inhibitor.

**Figure 3 cells-10-01772-f003:**
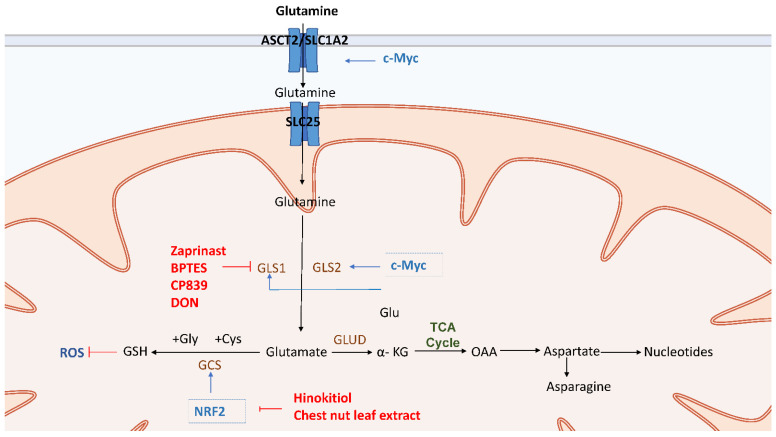
Glutaminolysis provides precursors for the biosynthesis of nucleic acids, lipids, amino acids and glutathione. Glutamine is obtained by tissue cells through ASCT2 (SLC1A5) and transformed into glutamate through glutaminase GLS1 and GLS2. Glutamate is then metabolized by glutamate dehydrogenase (GLUD) or by transaminase to generate α-KG, which serves as an anaplerotic substrate in the TCA cycle. Oncogene c-MYC induces overexpression of glutamine importers ASCT2 and GLS transcription. Nuclear factor-erythroid 2-related factor 2 (Nrf2) is a transcriptional regulator that binds to anti-oxidative genes promoter to maintain cellular redox status by promoting the expression of antioxidant enzymes such as gamma-glutamyl cysteine synthetase (GCS), an enzyme that generates GSH. Multiple therapeutic targets for GLS in CSCs glutaminolysis. Bis-2-(5-phenylacetamido-1,3,4-thiadiazol-2-yl) ethyl sulfide (BPTES) and CB-839 are allosteric inhibitors of GLS1, which inhibits the influx of glutamine derivatives into the TCA cycle. Zaprinast is a phosphodiesterase 5 inhibitor and an asthma drug that could effectively sensitize cancer stem cells and increase apoptosis through intracellular ROS accumulation in combination with Zaprinast and BPTES. Lastly, azaserine, acivicin, and 6-diazo-5-oxo-L-norleucine (DON) are glutamine analogs that inhibit the activity of GLS1. Nrf2 inhibitors are another potential therapeutic agent for CSCs. Hinokitiol has recently been discovered to suppress Nrf2 expression in glioma CSCs, resulting in decreased self-renewal capacity, migration, invasiveness, and colony-forming ability. In addition, treatment with chestnut leaf extract blocked the activation of the Nrf2 antioxidant protection in MCF-7-derived breast CSCs. Combination therapy of *Castanea crenata* leaf extract with paclitaxel could dramatically increase the death of cancer cells.

**Figure 4 cells-10-01772-f004:**
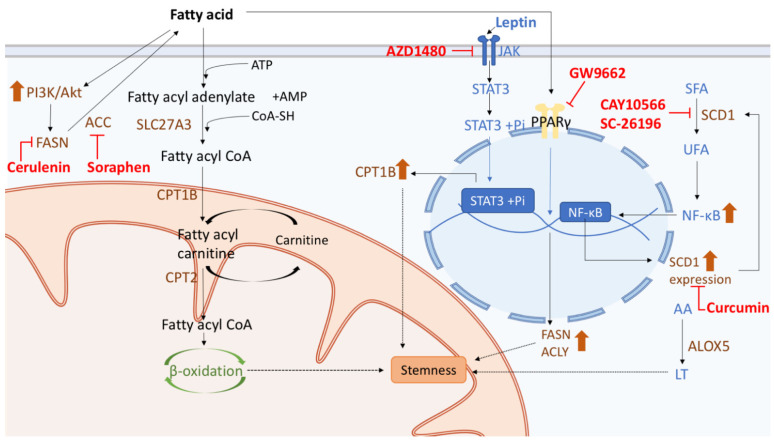
The regulation of fatty acid metabolism in CSCs. Lipid metabolism plays an important role in regulating the stemness properties of CSCs. In the pathway of fatty acid metabolism, acyl-CoA synthetase very-long-chain 3 (ACSVL3/SLC27A3) promotes the conversion of fatty acyl adenylate to fatty acyl CoA, which subsequently leads to β-oxidation of fatty acids and confers the stemness properties to CSCs. The generation of fatty acyl carnitine is supported by carnitine palmitoyltransferase 1B (CPT1B), which is upregulated by leptin-induced JAK/STAT3 signaling. JAK/STAT3 signaling can be repressed by AZD1480, a pan-JAK small molecule inhibitor, which may also affect the generation of CPT1B. Lipid desaturation has also been suggested to enhance CSCs maintenance, which is critically mediated by an increase of the stearoyl-CoA desaturase 1 (SCD1) activity. The enhanced expression of SCD1 leads to the transformation of saturated fatty acids (SFAs) to monounsaturated fatty acids (MUFAs), which subsequently promotes the activity of nuclear factor kappa B (NFκB). There is a feedforward loop that unsaturated fatty acids increase SCD1 via NFκB. The addition of curcumin blocks the expression of SCD1, which at last exerts the effect similar to CAY10566 and SC-26196 that influences the conversion of SFAs to MUFAs. Lipid metabolism enzymes such as arachidonic acid 5-lipoxygenase (ALOX5) also support the maintenance of stemness of CSCs by increasing the generation of leukotrienes (LT) from arachidonic acid (AA). Anticancer therapies targeting fatty acid metabolism may inhibit enzymes to modulate the stemness property of CSCs. The abovementioned pathways including fatty acid synthesis can be inhibited by cerulenin, soraphen and by blocking PPARγ signaling with GW9662, respectively.

## Data Availability

No new data were created or analyzed in this study. Data sharing is not applicable to this article.

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
