# Peer review of "Potential Therapies Targeting Metabolic Pathways in Cancer Stem Cells"

_cells, 2021, doi:10.3390/cells10071772_

Round 1

Reviewer 1 Report

The authors reviewed the unique metabolic pathways of cancer stem cells including mitochondrial respiration, glycolysis, glutaminolysis, and fatty acid metabolism. They highlight potential targets in these metabolic pathways for the development of more effective and alternative strategies to eradicate CCSs. They highlighted therapies targeting those specific pathways. This review is well detailed and very interesting.

Minor corrections

  1. Figure 1: Do you mean cytochrome c instead of cytokine C? Please correct.
  2. Please include the references of the research papers from which all figures were adapted.
  3. Line 254 to 256: please describe the results of the studies combining Metformin + PI3k inhibitor and metformin+ JQ1

Author Response

We appreciate and thank the editor and the reviewers for the constructive comments and suggestions to improve our manuscript. Our responses to the reviewers’ comments are summarized below. Our corrections are marked in blue to highlight the changes that we have made in our revised manuscript.

1. Comment # 1: Figure 1: Do you mean cytochrome c instead of cytokine C? Please correct.

Response: We sincerely appreciate the reviewer’s correction and we have already revised the Figure 1.

2. Comment # 2: Please include the references of the research papers from which all figures were adapted.

Response: We appreciate the reminder and we have included all the research papers that our figures have adapted.

3. Comment # 3: Line 254 to 256: please describe the results of the studies combining metformin + PI3k inhibitor and metformin+ JQ1.

Response: In section 2.3, we have removed the paragraph describing the combinatorial therapy of metformin and PI3k because the treatment targeted cancer cells rather than cancer stem cells, which we believe has little relevance to the topic of this review. We have moved the other paragraph describing combined regimen of metformin and JQ1 from the section of “glycolysis” to “mitochondrial respiration” since JQ1 enforced the dependence of cancer stem cells on OXPHOS and thus increased the effect of metformin on inhibiting mitochondrial respiration instead of glycolysis. This paragraph has been rewritten as “Finally, as a result of the metabolic plasticity of CSCs, BRD4 inhibitor JQ-1 enforced dependence on OXPHOS in pancreatic CSCs and deceased the intermediate glycolytic/respiratory phenotype, which enhanced the efficacy of metformin as an anti-mitochondrial respiration regimen [35]. This has enlightened us with the potential of the combinatorial therapies that impinge on adaptive metabolism of CSCs to subvert drug resistance and enhance the therapeutic efficacy of metformin, which is a clinically feasible approach.”

Reviewer 2 Report

Yao-An Shen and Chang-Cyuan Chen et al in this ms reviewed the distinct metabolic pathways in cancer stem cells (CSCs), focusing on mitochondrial respiration, glycolysis, glutaminolysis, and fatty acid metabolism, as well as the potential targeted cancer therapies. The review is generally well organized. The author should also discuss that the potential drug resistance from targeting one type of metabolism and the potential solutions. In addition, each metabolic regulation of CSC should also emphasize the closely linked oncogene/ tumor suppressor's functions. Minoe points:

  1. Line 132-134: "Metformin is a widely used anti-diabetic drug that can selectively target CSCs to interfere their cell function and survival without affecting their non-CSC counterparts." Metformin has many functions and the author mentions that it inhibits complex I, so how does metformin only inhibit complex I of  CSCs, but not that of non-CSC cells?
  2. Since mitochondrial respiration and glycolysis are closed associated, the authors stated that line 73-75 "Growing shreds of evidence have substantiated that most CSCs exhibit higher mitochondrial activity and preferentially use respiration and OXPHOS as the source of energy as compared with their non-CSCs counterpart (Fig. 1), which is exemplified in CSCs 75 isolated from ovarian cancer patients [16]." and line 179-181: "This phenomenon has been observed in many types of cancer, such as breast cancer, lung cancer, ovarian cancer, colon cancer, and even in osteosarcoma and glioblastoma [23,27,28,63,64]." The statement is a little confusing and the authors should further discuss the properties of combined glycolysis and mitochondrial activity status of CSCs vs non-CACs, instead of only focusing only one of them.

Author Response

We appreciate and thank the editor and the reviewers for the constructive comments and suggestions to improve our manuscript. Our responses to the reviewers’ comments are summarized below. Our corrections are marked in blue to highlight the changes that we have made in our revised manuscript.

1. Comment # 1: Line 132-134: "Metformin is a widely used anti-diabetic drug that can selectively target CSCs to interfere their cell function and survival without affecting their non-CSC counterparts." Metformin has many functions and the author mentions that it inhibits complex I, so how does metformin only inhibit complex I of CSCs, but not that of non-CSC cells?

Response: The statement that metformin can selectively target CSCs but not non-CSC counterparts was concluded based on the data but not fully explained by the original authors of the cited paper. After consideration of the reviewer’s comments, we have decided to remove the statement.

Metformin blocks mitochondrial respiration by targeting Complex I, which leads to a decrease in ATP production. This reduction of ATP induces activation of AMP-activated protein kinase (AMPK) pathway, a cellular energy sensor that coordinates metabolic pathways in many tissues. However, the mechanism underlying AMPK-mediated adaptation of cancer stem cells to survive warrants further clarification.

2. Comment # 2: Since mitochondrial respiration and glycolysis are closed associated, the authors stated that line 73-75 "Growing shreds of evidence have substantiated that most CSCs exhibit higher mitochondrial activity and preferentially use respiration and OXPHOS as the source of energy as compared with their non-CSCs counterpart (Fig. 1), which is exemplified in CSCs 75 isolated from ovarian cancer patients [16]." and line 179-181: "This phenomenon has been observed in many types of cancer, such as breast cancer, lung cancer, ovarian cancer, colon cancer, and even in osteosarcoma and glioblastoma [23,27,28,63,64]." The statement is a little confusing and the authors should further discuss the properties of combined glycolysis and mitochondrial activity status of CSCs vs non-CACs, instead of only focusing only one of them.

Response: The metabolic profile of ovarian cancer cells and many other kinds of cancer cells mentioned in the question above may switch between glycolysis and OXPHOS because of the variations in tumor microenvironment, external stress, and even in different cell lines. Also, the different metabolic preference mentioned in different papers about ovarian cancer stem cells differ because the ways how these cancer stem cells were isolated are not well defined so far. We have added a sentence in Section 2.1, “Like stem cells, CSCs with specialized adaptation permit metabolic switch between glycolysis and respiration to meet the cellular energy demand in response to changes of physiological conditions or microenvironmental stress such as quiescent state, low-oxygen concentration and nutrient deprivation.”

Reviewer 3 Report

This is a comprehensive review that collates a very wide range of observations related to metabolism in cancer (stem) cells. The inclusion of published results pertaining to all major metabolic pathways and a wide range of (mostly solid) cancers has resulted in what amounts to a catalogue of reported interactions and interventions of potential relevance. This is both a major strength and a major weakness of the review, since the scope of potentially relevant metabolic pathways and modulators is so extensive that the coverage of each issue has to be very superficial indeed. There is no attempt (and no space) to consider underlying issues in any detail and therefore no real discussion or consideration possible.

Although the authors acknowledge at the beginning and the end of their review that any therapy will have to be targeted to properties specific to cancer stem cells as opposed to the normal ones that are essential for tissue homeostasis, this issue is not given sufficient attention. The diversity of studies referred to in this review employ and address cell populations that are variously defined, implied or simply enriched in cancer stem cells. There is no clear explanation of which of the pathways considered may be particularly specific to stem cells as opposed to the bulk of their respective Tumors,or why this should be the case. A clearer explanation of how the properties of cancer stem cells may be reflected in specific patterns of metabolism that are not found in their normal counterparts would be most helpful. Without this, one could almost remove the “Stem” from the title.

Author Response

We appreciate and thank the editor and the reviewers for the constructive comments and suggestions to improve our manuscript. Our responses to the reviewers’ comments are summarized below. Our corrections are marked in blue to highlight the changes that we have made in our revised manuscript.

-------

Comments and Suggestions for Authors: Although the authors acknowledge at the beginning and the end of their review that any therapy will have to be targeted to properties specific to cancer stem cells as opposed to the normal ones that are essential for tissue homeostasis, this issue is not given sufficient attention. The diversity of studies referred to in this review employ and address cell populations that are variously defined, implied or simply enriched in cancer stem cells. There is no clear explanation of which of the pathways considered may be particularly specific to stem cells as opposed to the bulk of their respective tumors, or why this should be the case. A clearer explanation of how the properties of cancer stem cells may be reflected in specific patterns of metabolism that are not found in their normal counterparts would be most helpful. Without this, one could almost remove the “Stem” from the title.

Response: We appreciate the critical comments of the reviewer and have made proper revisions to address this important issue. The rationale behind this review is that the metabolic pathway in cancer stem cells diversifies and switches according to different microenvironments and external stress but that in the bulk of cancer cells prefers glycolysis, known as Warburg’s effect. The capability of switching among different metabolic pathway, known as metabolic plasticity, rather than a specific metabolic pattern separate cancer stem cells from the bulk of tumor cells. We envision that targeting metabolic pathways in cancer stem cells can sensitize the rest of tumor cells to conventional therapy and reduce the recurrence of tumor by hindering the metabolism of cancer cells and cancer stem cells simultaneously. Indeed, controversies remains since there is no clear and consensus definition of cancer stem cells by far. Moreover, the isolation of cancer stem cells / stem-like cancer cells varies in same cancer type or even in same cell lines, which leads to generation of different metabolic phenotype of cancer stem cells. The value of this review article is to summarize the four main metabolic pathways currently discussed in cancer stem cells and the potential corresponding anti-metabolic regimens.

Accordingly, we have revised and added several paragraphs in the Introduction section to highlight the reasons why we have aimed to explore the therapies aimed to CSCs but not their respective tumors, and also put more emphasis on the reasons why we discuss about the metabolism in CSCs rather than other characteristics and markers of CSCs. The added paragraphs mainly describe the differences between CSCs and their tumors, including their surface cell markers, functional properties, and expression of specific genes, and the metabolic plasticity, which have been previously suggested as therapeutic targets. Finally, we have presented an alternative approach to identify CSCs by their metabolic profiles, which are distinct from those of cancer cells, and put emphasis on the potential of eradicating cancer cells by targeting to CSC metabolic pathways.

Round 2

Reviewer 3 Report

The modifications made by the authors have helped to put their work in context and thus improved the accesibility of their review.

Author Response

We thank the reviewers for their constructive comments and suggestions to improve the content of our manuscript after appropriate revisions.